# Comparative Transcriptome Profiling Analysis Uncovers Novel Heterosis-Related Candidate Genes Associated with Muscular Endurance in Mules

**DOI:** 10.3390/ani10060980

**Published:** 2020-06-04

**Authors:** Shan Gao, Hojjat Asadollahpour Nanaei, Bin Wei, Yu Wang, Xihong Wang, Zongjun Li, Xuelei Dai, Zhichao Wang, Yu Jiang, Junjie Shao

**Affiliations:** College of Animal Science and Technology, Northwest A&F University, Yangling, Xianyang 712100, China; gaoshan@nwafu.edu.cn (S.G.); h.asadollahpour1988@gmail.com (H.A.N.); wei_bin1012@163.com (B.W.); wang_yu@nwafu.edu.cn (Y.W.); rebecca_830410@163.com (X.W.); lizongjun@nwsuaf.edu.cn (Z.L.); daixuelei2014@163.com (X.D.); wangzhichao1128@163.com (Z.W.); yu.jiang@nwsuaf.edu.cn (Y.J.)

**Keywords:** heterosis, mule, alternative splicing, differently expressed genes

## Abstract

**Simple Summary:**

Mules have better and greater muscle endurance than hinnies and their parents. However, the molecular mechanisms underlying heterosis in their muscles are still much less understood. In this study, we conducted comparative transcriptome and alternative splicing analysis on the heterosis mechanism of muscular endurance in mules. Our results showed that 8 genes were significantly enriched in the “muscle contraction” pathway. In addition, 68% of the genes with alternative splicing events from the mule muscle tissue were validated by the long transcript reads generated from PacBio sequencing platform. Our findings provide a research foundation for studying the genetic basis of heterosis in mules.

**Abstract:**

Heterosis has been widely exploited in animal and plant breeding programs to enhance the productive traits of hybrid progeny from two breeds or species. However, its underlying genetic mechanisms remain enigmatic. Transcriptome profiling analysis can be used as a method for exploring the mechanism of heterosis. Here, we performed genome-wide gene expression and alternative splicing (AS) analyses in different tissues (muscle, brain, and skin) from crosses between donkeys and horses. Our results indicated that 86.1% of the differentially expressed genes (DEGs) and 87.2% of the differential alternative splicing (DAS) genes showed over-dominance and dominance in muscle. Further analysis showed that the “muscle contraction” pathway was significantly enriched for both the DEGs and DAS genes in mule muscle tissue. Taken together, these DEGs and DAS genes could provide an index for future studies of the genetic and molecular mechanism of heterosis in the hybrids of donkey and horse.

## 1. Introduction

Heterosis, also known as hybrid vigor, refers to the phenomenon that hybrids exhibit superior performance in areas such as stress resistance, fertility, growth rate, and biomass production compared with their parental inbred lines [1,2,3]. This phenomenon has been widely used in breeding programs, by mating two different pure-bred lines, to improve the quantity of both crops [4,5,6] and livestock production [7,8,9]. There are three main hypotheses to explain the genetic mechanism of heterosis: dominance [10,11], over-dominance [12], and epistasis [13]. The dominance model proposes that the actions of deleterious recessive alleles are suppressed by a dominant homolog. The over-dominance model defines that allelic interactions at a single locus or more loci lead to increased vigor. Unlike these two models, the third model explains interactions between nonallelic genes, which are created due to new combinations in the hybrid. With the development of functional genomics and the improvement of RNA sequencing (RNA-seq) technologies, the molecular basis of heterosis has been investigated at the transcriptional level of gene expression in several plant and animal species such as cattle [14], mice [15], pig [16], chicken [17], coral reef butterflyfish [18], and crossed lamb [19]. 

Alternative splicing (AS) is a process during gene expression that can produce various mature mRNAs encoding different proteins (protein isoforms) from a single gene [20]. Furthermore, the varied proportion of different splicing isoforms can affect and alter gene expression, which may also contribute to heterosis. For example, two different transcriptome isoforms of the *Titin* gene are expressed in different stages of heart development: the predominant expression of the long isoform (N2BA) is required during the development stage, while the expression of the shorter one (N2B) is exclusively dominant in adults [21,22,23]. Thus, if the N2BA isoform is expressed mainly in the adult heart, it can cause heart disease [24,25]. 

In the past decade, advances in RNA-Seq technology have provided an opportunity to investigate the molecular aspects of heterosis in many plants and animals; however, the expressions of heterosis-related genes in mules and hinnies (crosses between donkeys and horses) have not been determined clearly. Mules, which normally exhibit hybrid vigor in their physical characteristics, are renowned for having better and greater muscle endurance than hinnies and their parents [26,27,28]. Apart from skeletal muscle tissue, which is the main phenotypic differencing mules from hinnies and their parents, there are also clear differences between these animals in both brain and skin. For example, among all equine species, horses have often not performed well in learning tasks; however, mules are better than donkeys in the visual discrimination learning task [26,28]. On the other hand, the donkey’s skin is well-adapted to hot dry climates but is much less water-resistant than the horse’s skin and can easily become saturated with rain, leading to skin maceration [29]. In this study, in order to explore the genetic and molecular mechanisms of heterosis-related genes in mules, we identified differentially expressed genes (DEGs) and differential alternative splicing (DAS) events in tissues from muscle, brain, and skin. Analysis of gene expression profiling and AS events in the examined tissues revealed significant differences between mules and hinnies, as well as between mules and both of their parents. The findings provide new insights into the genetic mechanism underlying heterosis in mules.

## 2. Materials and Methods

### 2.1. Sample Collection 

In this study, the experimental protocol was approved by the Institutional Animal Care and Use Committee of Northwest A&F University (Permit Number: NWAFAC1019). All hybrid samples from horse and donkey (10-year-old) were collected from Yulin City, Shaanxi Province. Tissue samples, including muscle (semitendinosus muscle, n = 7, four hinny and three mule samples), brain (prefrontal lobe, n = 5, two hinny and three mule samples) and skin (back of neck, n = 7, four hinny and three mule samples) were dissected and rinsed with PBS.

### 2.2. RNA Preparation and Single-Molecule Sequencing

All short reads sequencing data used for this study were obtained from our previous project [30]. The RNA samples used for generating Pac-Bio long reads were extracted from the same muscle tissue from mule (number: M2M) as in our previous project [30]. Total RNA was extracted using Trizol solution following the manufacturer’s instructions. RNA concentration and quantification for semitendinosus muscle tissue were done using the NanoDrop 2000 spectrophotometer (Thermo Scientific, Waltham, MA, USA) and Agilent Bioanalyzer 2100, respectively. Five different size fragment sequencing libraries (1–2 kb, 3 × 2–3 kb, >3 kb) were constructed according to the guide for preparing the Single-Molecule Sequencing in Real Time (SMRT) bell template for sequencing on the PacBio RS platform. In addition, horse RNA-Seq data were downloaded from the Natinonal Center for Biotechnology Information (NCBI) accession numbers PRJNA339185 [31] and PRJEB26787 [32]. Moreover, in this study, all raw sequencing data generated from the PacBio sequencing platform were uploaded to NCBI (accession number: PRJNA560325).

### 2.3. RNA Reads Trimming and Alignment

The adapter sequences were first removed from all RNA-seq raw reads. Then, all reads were trimmed using Trimmomatic (v0.33, Aachen, Germany) [33]. Further, those reads longer than 70 nucleotides in length were retained as high-quality clean data. Finally, the high-quality reads from mule/hinny and their parents were aligned to the horse reference genome (Equus caballus: EquCab2.0) by STAR (v 2.5.1a, New York, NY, USA) [34]. 

### 2.4. Differential Expression Analysis

Once the reads were mapped to the reference genome, the read quantification was performed using Stringtie (v. 1.3.4d, Baltimore, MD, USA) [35,36] and the attached python script prepDE.py (http://ccb.jhu.edu/software/stringtie/index.shtml?t=manual). The number of DEGs was calculated using the DESeq2 R package (v. 1.10.1, Heidelberg, Germany) [37]. Finally, the corrected p-values under 0.05 were deemed significant DEGs.

### 2.5. Functional Enrichments Analysis

The Kyoto Encyclopedia of Genes and Genomes (KEGG) pathways enrichment analysis was carried out using the online KOBAS (v. 3.0, Beijing, China) annotation tool (http://kobas.cbi.pku.edu.cn/) [38]. The protein sequences derived from DAS and DEGs genes were submitted to the human database, and hypergeometric test/Fisher’s exact test were used to calculate the corrected *p*-values.

### 2.6. Analysis of DAS Genes 

A replicate multivariate analysis of transcript splicing (rMATs, v. 3.2.5, Philadelphia, PA, USA) [39] was applied for comparison and identification of DAS events, including exon skipping (SE), mutually exclusive exons (MXE), alternative 5’ splice site (A5SS), alternative 3’ splice site (A3SS), and retained intron (RI). To test the significance of rMATs, the likelihood-ratio method was implemented by calculating the *p*-value based on the differential ψ values, also known as “percent spliced-in” (PSI). The splicing events were assessed with rigorous statistical criteria (|Δψ|> 10% and FDR ≤ 0.05), which quantified AS.

### 2.7. PacBio Full-Length Transcripts Analysis

The PacBio RS SMRT sequence reads were first analyzed on the Pacific Biosciences’ SMRT analysis software (v. 2.3.0, Silicon Valley, CA, USA) to get reads of insert (ROI). Then, the ROIs were aligned to the corresponding reference genome using GMAP (v. 2015-12-31, San Francisco, CA, USA) [40]. The high error rate of long reads was corrected using TAPIS (https://bitbucket.org/comp_bio/tapis). Then, DAS events were visualized using the R package Gviz (v. 1.18.1, Basel, Switzerland) [41].

## 3. Results

### 3.1. Gene Expression Level in Muscle, Brain, and Skin Tissues of Mule

Our results showed that more than of 601, 427, and 482 million clean reads were retained from muscle, brain, and skin tissues, respectively, after removing the adaptor and low-quality reads. Moreover, the average mapped reads ratios with reference genome were 91.50%, 90.22%, and 91.81% for the above three tissues, respectively. The analysis of gene expression correlation indicated that all samples were firstly clustered by tissues and thereafter by species, respectively (Figure 1a). A similar pattern was also shown by principal component analysis (PCA) (Appendix A).

### 3.2. Analysis of DEGs between Hybrids and Either of Their Parents

When comparing samples from mule and horse, a total of 7044, 3689, and 6637 DEGs (adjusted *p* <  0.05, log_2_ transformed > 1) were identified from muscle, brain, and skin tissues, respectively. Among them, 3442, 2162, and 3560 of DEGs were up-regulated in the above three tissues, while 3602, 1527, and 3077 of DEGs were down-regulated, respectively (Table 1). On the other hand, when comparing samples from mule and donkey, 275, 952, and 1326 DEGs (adjusted *p* <  0.05) were identified from muscle, brain, and skin tissues, respectively. A total of 202, 611, and 606 DEGs were up-regulated, while 73, 341, and 720 DEGs were down-regulated, respectively (Table 1). It is not surprising that the number of identified DEGs between mule and horse are higher than between mule and donkey. First, the horse RNA-Seq data were obtained from an adult thoroughbred of a different age; however, RNA-Seq data for donkey tissues were obtained from our previous project: the mule and donkey were exposed to the same growing environment. Another reason might be attributable to different library preparation procedures for generating sequencing data. In addition, a number of DEGs between the hinny and either of the parents were also detected (Table 1 and Appendix A). Then, we divided identified DEGs from both mules and their parents into two groups: additive and non-additive patterns. We found 7196, 4334, and 7367 DEGs (adjusted *p*  < 0.05) (Table 2, Appendix A) from muscle, brain, and skin tissues between mule and either of their parents, respectively. Among these genes, a total of 6762, 3709, and 6144 DEGs showed a non-additive pattern in the mentioned tissues, respectively (Table 2). Also, 3764, 1306, and 2535 DEGs exhibited non-additive gene actions of over-dominance and under-dominance in the above three tissues, while 2437, 1715, and 2601 DEGs showed non-additive gene actions of high-parent dominance and low-parent dominance modes, respectively (Table 2). Based on the hierarchical clustering algorithm, the expression patterns of DEGs in all three mule tissues were more closely related to the donkey than the horse (Figure 1b); however, the expression patterns of DEGs in those tissues were irrelative between the hinny and either of the parents. To further investigate the biological function of DEGs in hybrids, a pathway enrichment analysis of the DEGs was conducted using KOBAS (Appendix A). The identified DEGs from all hybrid tissues were significantly enriched in the “muscle contraction”, “neuronal system”, and “DNA repair” pathways.

### 3.3. Identification and Characterization of DAS Events

A total of 2241, 2657, and 1973 DAS events were identified from 1240, 1674, and 1330 genes in mule muscle, brain, and skin tissues, respectively (Appendix A). Based on the gene AS analysis, all samples were clustered first by tissue type (PSI value) and then by species (Figure 2a), which is consistent with the results of gene expression analysis. Similarly, hinny DAS genes were detected in all three tissues as mentioned above (Table 3). The classification of DAS genes was the same as mentioned above regarding DEGs (Figure 2b and Table 4). As depicted in Table 4, the highest proportion of DAS genes in all three tissues showed high-parent dominance pattern (950, 42.4%), followed by low-parent dominance (633, 28.2%), under-dominance (198, 8.8%), and over-dominance (173, 7.7%), respectively. The results of gene enrichment analysis from mule DAS genes showed that the “muscle contraction” and “neuronal system” pathways are significantly enriched in the muscle and brain tissues, respectively (Appendix A). Genes that exhibited an over-dominance pattern were significantly enriched in the pathways of “vasopressin synthesis” and “circadian rhythm”, while genes that showed a dominance pattern were significantly enriched in the “striated muscle and contraction” and “muscle contraction” pathways (Figure 2c,d).

### 3.4. DEGs and DAS Genes in Muscle Contraction Pathway

By combining the results of comparison analysis between mule and hinny, we found 5.8–11.2%, 3.4–10.9%, and 4.0–12.6% of DEGs with DAS from muscle, brain, and skin tissues, respectively (Appendix A). On the other hand, 68% of DAS genes identified in the mule muscle tissue were verified by the long reads generated from the PacBio sequencing platform (Table 5 and Appendix A). Further analyses revealed that those DEGs and DAS genes identified from muscle tissue were significantly enriched in the “muscle contraction” pathway (Figure 3). A total of 68 genes were mapped in this pathway. Among them, 42 and 18 were identified in the DEGs and DAS genes, respectively. Moreover, 8 genes (*TNNC2*, *RYR1*, *STIM1*, *CAMK2D*, *CAMK2B*, *CACNA1S*, *DYSF*, and *ATP2A1*), were shared by both DEGs and DAS genes. In the hybrid individuals and their parents, the *TNNC2* gene was mainly expressed in the fast skeletal muscle, and its expression level was two times higher in the mule than that in the horse (Figure 4a). 

In addition, one SE event was identified in this gene, located on the chr22:34802038-34802136, when its transcriptome isoform was mainly expressed in the mule muscle (Figure 4b). Also, the *RYR1* gene was detected in this pathway, and its expression level was higher in the skeletal muscle samples than in other tissues (Figure 4c). The expression level of this gene was more than two times higher in mule muscle that in horse muscle. This gene is located on the chromosome 10 and contained a SE event in position chr10:9571134-9571148 (Figure 4d and Appendix A). The SE event of the *RYR1* gene trends to skip in the horse muscle genome, while it tends to remain in the mule muscle genome (Figure 4d).

## 4. Discussion

The mule is one of the most common and important conveyance means for people who living in mountainous areas, owing to their outstanding performance in muscular endurance in comparison with its parents [26,27,28]. However, the genetic and molecular mechanisms of heterosis in mules remain unknown. Based on the results from the correlation of gene expression level and AS events, the divergence between hybrids and either of their parents can be indicated by DEGs and DAS genes.

### 4.1. Comparative Transcriptome Analysis Between Hybrids and Their Parents

By comparing transcriptomic data collected from hybrid individuals and their parents, we identified a subset of DEGs in brain, muscle, and skin tissues. The correlations of gene expression level and genes with AS events were first clustered by tissue and then by species, which showed a high relative reliability of samples (Figure 1a). The gene expression profiles of tissue samples from both mule and donkey were clustered together. Also, our results showed that the gene expression profiles of brain tissue from both hinny and horse were grouped together. Similar results were also observed in the DAS genes. The overall findings suggest that although the genetic materials for all hybrid individuals were the same, a great variety at the level of gene expression was found between mules and hinnies. Moreover, the DEGs mainly showed over-dominance, while the majority of the DAS genes exhibited dominance models. Differences in gene expression profiles and AS events between mule and hinny may be caused by the genetic basis of heterosis. 

### 4.2. Differential Alternative Splicing Contributes to Heterosis of Hybrids

Through comparative transcriptome analysis, we found that all identified DAS genes from different tissues belonged to the same species. Moreover, according to Pearson’s correlation coefficient, we found that the correlation matrix of DAS genes is more relevant in hybrid individuals and either of their parents. These results suggest that the differential splicing events between hybrids and either of their parents may arise from heterosis. In addition, we found several common genes between DEGs and DAS in skin tissues, which were enriched in the “DNA repair” pathway (Appendix A). Among these genes, we found the cyclin-dependent kinase 7 (*CDK7*) gene, which is a ubiquitous kinase and regulates key events in cell cycle progression and transcription [42]. Previous research studies showed that this gene has an important role in highly proliferative tissues, such as skin and intestine [43]. We also observed that the DEGs of brain tissues were significantly enriched in the “neuronal system” (corrected *p* = 6.66 × 10^−6^) and “axon guidance” (corrected *p* = 1.32 × 10^−26^) pathways. In addition, we found the *GRM5* (metabotropic glutamate receptor 5) gene, which encodes a glutamate metabotropic receptor (Appendix A). It has been showed that that genetic mutations in *GRM5* are associated with cognitive impairments and right hippocampal volume reduction in schizophrenia. Moreover, it was demonstrated that the receptor encoded by this gene is needed for normal brain function [44,45]. 

### 4.3. Muscle Contraction Pathway in Heterosis

By combining the results of KEGG enrichments analysis of DEGs and DAS genes, we found 8 common genes that were significantly enriched in the “muscle contraction” pathway. Among them, *TNNC2* and *RYR1* exhibited a dominance pattern in mule muscle tissue. Moreover, the proportion of transcriptome isoforms of these two genes are varied when they are expressed in mule and horse muscle tissues. The encoded protein of the *TNNC2* gene plays an important role in overcoming the inhibitory effect of the troponin complex on actin filaments [46,47]. In addition, this gene acts as a calcium release channel in the sarcoplasmic reticulum and is a connection between the sarcoplasmic reticulum and the transverse tube [48,49]. The *RYR1* gene plays a signal role in embryonic skeletal muscle formation. This gene is mainly expressed in the sarcoplasmic reticulum (SR) of skeletal muscle and serves as the major Ca^2+^ release channel required for cell contraction. Furthermore, previous studies showed that there is a correlation between the *RYR1* gene and fiber size and structure, as well as fiber type predominance, in muscle [50]. Recent horse genome studies also have discovered genes involved in skeletal muscle development and function. For example, Ablondi et al. (2019) [51] identified loci related to muscle contraction and function in sport horses. In another study, Asadollahpour Nanaei et al. (2019) [52], using whole genome resequencing data, revealed genes and signaling pathways involved in physical and athletic performance in Hanoverian horses. In brief, our results indicated that AS events are linked to heterosis in hybrid individuals. Further experimental verification will be required in order to reveal the regulatory mechanism of mRNA splicing on gene regulation in hybrids and their parents.

## 5. Conclusions

Our results showed that the “muscle contraction” pathway is significantly enriched by both the DEGs and DAS genes. We also found some candidate genes involved in different biological and cellular functions including those affecting muscular endurance traits in mule. Our study provides valuable resources for further research on the genetic and biological basis of heterosis in mules.

## Figures and Tables

**Figure 1 animals-10-00980-f001:**
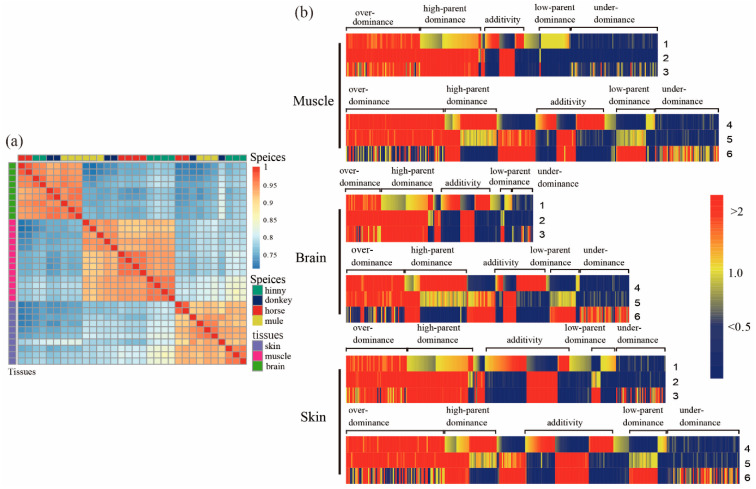
Differentially expressed genes (DEGs) in tissues of muscle, brain, and skin from mule, hinny, and their parents. (**a**) Pearson’s correlation of the expression profile of genes in the tissues of brain, muscle, and skin from mule, hinny, and their parents. The clustered results of gene expression among species are shown on the top, while the clustered results of gene expression in these tissues of species are shown on the right side. (**b**) Hierarchical clustering display of differentially expressed genes with the ratio of intensity according to expression patterns of genes in mule, hinny, and their parents. The color scale is shown at the top, and the mode of gene action is on the side. Lane 1 represents intensity ratios of mule to horse; Lane 2, intensity ratios of mule to donkey; Lane 3, intensity ratios of horse to donkey; Lane 4, intensity ratios of hinny to horse; Lane 5, intensity ratios of hinny to donkey; Lane 6, intensity ratios of horse to donkey.

**Figure 2 animals-10-00980-f002:**
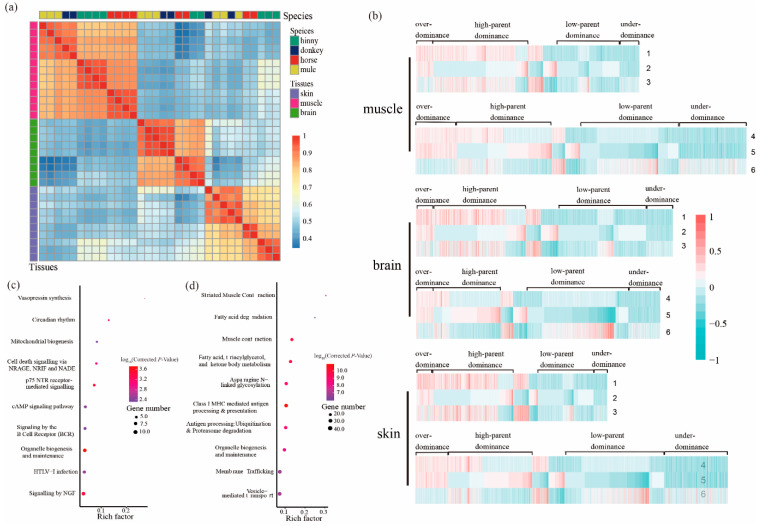
Genes with DAS events in the tissues of muscle, brain, and skin from mule, hinny, and their parents. (**a**) Pearson’s correlation coefficient of the PSI (percent spliced-in) of genes with DAS events in different tissues. The gene clustered results of DAS events among species are shown on the top of the plot, while the clustered results of genes with DAS events among tissues are shown on the right side of the plot. (**b**) Hierarchical clustering display of DAS genes with the ratio of intensity according to expression patterns for horse, donkey, mule, and hinny. The color scale is shown at the top and the mode of gene action is shown on the side. Lane 1 represents intensity ratios of mule to horse; Lane 2, intensity ratios of mule to donkey; Lane 3, intensity ratios of horse to donkey; Lane 4, intensity ratios of hinny to horse; Lane 5, intensity ratios of hinny to donkey; Lane 6, intensity ratios of horse to donkey. (**c**) Kyoto Encyclopedia of Genes and Genomes (KEGG) pathways enriched by the genes that follow the hypothesis of over-dominance. (**d**) KEGG pathways enriched by genes that follow the hypothesis of dominance.

**Figure 3 animals-10-00980-f003:**
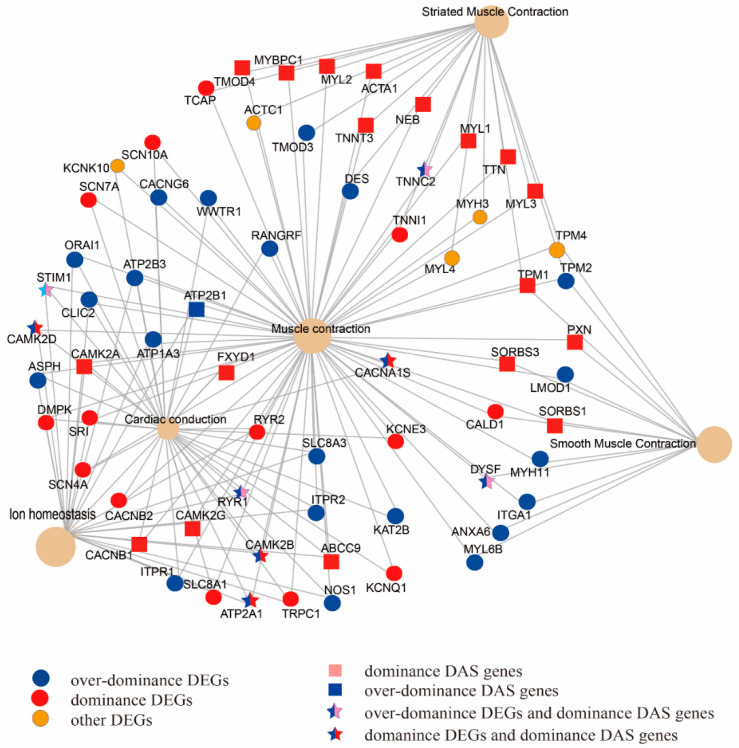
DEGs and genes with DAS events belonging to the models of over-dominance and dominance were enriched in the muscle contraction pathway.

**Figure 4 animals-10-00980-f004:**
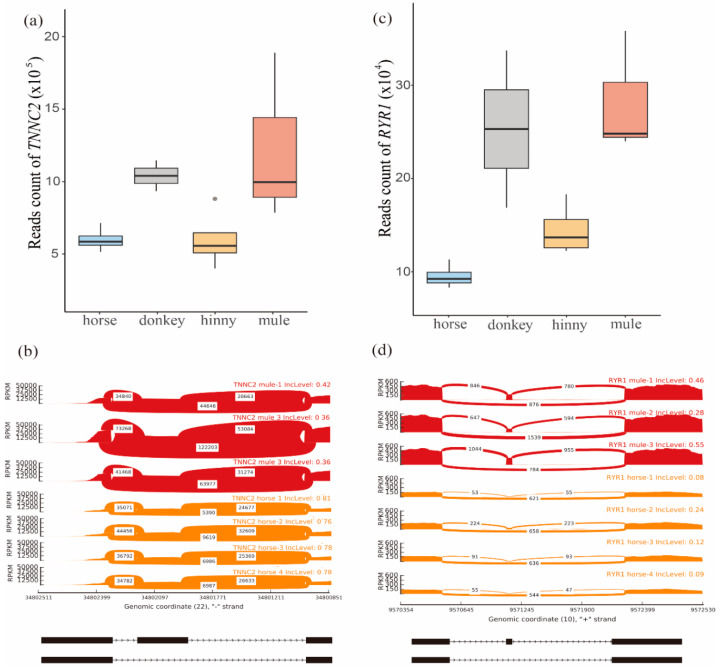
Analysis of *TNNC2* and *RYR1* genes. (**a**) Distribution of reads mapped on the *TNNC2* gene in muscle tissue of mule, hinny, and their parents. (**b**) Sashimi plot of the *TNNC2* gene showing DAS events in mule muscle tissue. (**c**) Distribution of reads mapped on the *RYR1* gene in muscle tissues of mule, hinny, and their parents. (**d**) Sashimi plot of *RYR1* gene showing DAS events in the mule muscle tissue. Read densities supporting inclusion and exclusion of exons are shown.

**Table 1 animals-10-00980-t001:** Distribution of DEGs across muscle, brain, and skin tissues of mule, hinny, and their parents.

Tissues	Gene Type	Mules vs. Horses	Mules vs. Donkeys	Hinnies vs. Horses	Hinnies vs. Horses
Muscle	Up-regulated	3602	73	2523	2442
	Down-regulated	3442	202	2810	3028
Brain	Up-regulated	1527	341	1054	2308
	Down-regulated	2162	611	1223	2895
Skin	Up-regulated	3077	720	2899	2573
	Down-regulated	3560	606	3311	2645

**Table 2 animals-10-00980-t002:** Classification of DEGs via gene expression patterns.

Tissues	Hybrids	Non-Addictive	Over-Dominance	High-Parent Dominance	Addictive	Low-Parent Dominance	Under-Dominance
Muscle	Mule	561	1713	1397	434	1040	2051
	Hinny	1135	2281	1207	1371	1160	1459
Brain	Mule	688	811	1212	625	503	495
	Hinny	992	1369	1431	826	792	1148
Skin	Mule	1008	1394	1532	1223	1069	1141
	Hinny	1363	2294	1189	1343	1237	1665

Hybrids are mule or hinny; P, paternal lines represent male horse or male donkey; M, maternal lines represent female horse or female donkey. Additivity, mule or hinny = 12 (horse + donkey); non-additivity, mule or hinny > 12 (horse + donkey) or mule or hinny < 12 (horse + donkey). High-parent dominance, mule or hinny = P > M or mule or hinny = M > P; low-parent dominance, mule or hinny = P < M or mule or hinny = M < P; over-dominance, mule or hinny > P and mule or hinny > M; under-dominance, mule or hinny < P and mule or hinny < M.

**Table 3 animals-10-00980-t003:** Distribution of genes with differential alternative splicing (DAS) events across the tissues of muscle, brain, and skin from mule, hinny, and their parents.

Tissues	Mules vs. Horses	Mules vs. Donkeys	Hinnies vs. Horses	Hinnies vs. Horses
Muscle	1377	593	656	1319
Brain	1124	280	1264	1330
Skin	1128	378	1121	1570

**Table 4 animals-10-00980-t004:** Classification of genes via differentially splicing events following the gene expression patterns.

Tissues	Hybrids	Over-Dominance	High-Parent Dominance	Addictive	Low-Parent Dominance	Under-Dominance
Muscle	Mule	173	950	287	633	198
	Hinny	450	1037	311	1075	744
Brain	Mule	189	946	334	910	278
	Hinny	175	695	278	1043	333
Skin	Mule	171	828	253	578	143
	Hinny	340	866	335	1033	655

Hybrids represent mule or hinny; P, paternal lines represent male horse or male donkey; M, maternal lines represent female horse or female donkey. Additivity, mule or hinny = 12 (horse + donkey); non-additivity, mule or hinny > 12 (horse + donkey) or mule or hinny < 12 (horse + donkey). High-parent dominance, mule or hinny = P > M or mule or hinny = M > P; low-parent dominance, mule or hinny = P < M or mule or hinny = M < P; over-dominance, mule or hinny > P and mule or hinny > M; under-dominance, mule or hinny < P and mule or hinny < M.

**Table 5 animals-10-00980-t005:** Results of genes with DAS events verified by the PacBio long reads.

Comparison Mule and Either of Its Parents	Covered DAS Genes	Verified DAS Genes	(Verified/Covered) %
mule vs. horse	1123	768	68
mule vs. donkey	279	183	66

## Data Availability

The RNA-seq data for this publication were submitted to the National Center for Biotechnology Information (https://www.ncbi.nlm.nih.gov/) and assigned the accession PRJNA560325.

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
