# Peer review of "Comparative Transcriptome Profiling Analysis Uncovers Novel Heterosis-Related Candidate Genes Associated with Muscular Endurance in Mules"

_animals, 2020, doi:10.3390/ani10060980_

Round 1
Reviewer 1 Report
This study discovered DEGs and DAS in Mule tissues across horse and donkeys. Pacbio long read seq platform was applied to get the RNA-seq data. Particularly, the bioinformatic methods for DAS analysis in this study are quite soundness and will be very interested to the readers of this journal. The authors led their results of DAS to interpret the heterosis of hybrids species. So, I recommend this manuscript would be good to be published after revision of my below comments.
- I wonder why this study used and focused on three tissues, Muscle, Brain and Skin. Is there any scientific reason to choose the tissues? Please add to mention it in the introduction.
- Discussion part has not been much narrow down to gene levels in Brain and Skin tissues, whereas the authors focused much in Muscle. Is there any reason it? If the authors could not discovered any meaningful results from the two tissues, I think it could be better to remove the whole materials about the two tissues in the revised manuscript. And the authors would be better to only focus on the Muscle tissue.
- In the materials and methods part, please put more details for sample collection process like which muscle and brain part were taken, what ages/month of animals were sacrificed, and other genetic background of the studied animals including gender and pedigree information as well.
Author Response
Reviewer
This study discovered DEGs and DAS in Mule tissues across horse and donkeys. Pacbio long read seq platform was applied to get the RNA-seq data. Particularly, the bioinformatic methods for DAS analysis in this study are quite soundness and will be very interested to the readers of this journal. The authors led their results of DAS to interpret the heterosis of hybrids species. So, I recommend this manuscript would be good to be published after revision of my below comments.
1. I wonder why this study used and focused on three tissues, Muscle, Brain and Skin. Is there any scientific reason to choose the tissues? Please add to mention it in the introduction.
Authors: Thanks for your valuable comment! We have added some details about the importance of these tissues in the introduction section. please see P3:L62-68.
2. Discussion part has not been much narrow down to gene levels in Brain and Skin tissues, whereas the authors focused much in Muscle. Is there any reason it? If the authors could not discovered any meaningful results from the two tissues, I think it could be better to remove the whole materials about the two tissues in the revised manuscript. And the authors would be better to only focus on the Muscle tissue.
Authors: Thanks for your comment! We have added some details in the discussion, please see P15:L330-347.
3. In the materials and methods part, please put more details for sample collection process like which muscle and brain part were taken, what ages/month of animals were sacrificed, and other genetic background of the studied animals including gender and pedigree information as well.
Authors: It has been revised. Please see P4:L83:87.
Reviewer 2 Report
Comparative transcriptome profiling analysis uncovers novel heterosis-related genes associated with muscular endurance in mule
The study is interesting, it is the continuation of another study of the same authors. Results show that despite expected on the heterosis of Mules and Hinnies, there is not yet published in the literature on the candidate genes presented here, paving the way for new studies.
Suggested changes to accept the publication are as follows:
TITLE
The authors conclude that there are candidate genes (line 285), it may be indicated to put the title “candidates genes”.
INTRODUCTION
A good explanation of the topic under study, short and objective. Authors should pay attention to:
Lines 42: The number of references to animals could be higher than that of Plants, since they refer to 2 and 7 references, respectively. Even though there are many more studies of plants, there are more than two studies of animals that can be cited.
Lines 51: Avoiding the terms "Although the recent" with the evolution and the passage of time, is not recent. Include an analogy to the moment (decade x, around year x, beginning of century x).
MATERIAL AND METHODS
It allows the repetition of the study, the Pac-Bio are correctly inserted NCBI.
The number (N) of animals evaluated in the study, despite appearing in the study by the same authors in
https://www.ncbi.nlm.nih.gov/pmc/articles/PMC6680129/ I think it should be inserted in the present study, obliging the reader to go to another study just to consult the N, it is not necessary.
Line 90: It is correct, but you can put the reference that indicates the use of the ideal P-value to be 0.05, when using the DESeq2 R package (v 1.10.1), or in the package manual, it indicates, if yes, you do not need to repeat the reference then.
RESULTS
Throughout the study, the authors repeat several times what they are assessing, will not be necessary.
Example: Phrase from line 112-113.
Line 116: it will be interesting to discuss and compare these values in the discussion section.
Lines 134 - 140 and serves for this whole paragraph: Table 1 is missing an indication at the end of the first sentence and review other text and see if the tables and figures are missing as soon as they speak the first time in the result.
They do not need to repeat the numerical values of the results of the tables, just put the percentages in the text, nor do they need to indicate the percentage differences of both, if one is 48.9% (line 134) logically that the other is 51.1% (line 135). Example as they can write: “Among them 48.9%, 58.6% and 53.6% of DEGs exhibited up-regulated in the above three tissues, respectively, while the other DEGs exhibited down-regulated (Table 1).”
Tables S1 and S2 can be joined.
Line 146: Correct the indication for the supplementary figure “… and Supplementary Figure 2” keeping the pattern they placed in the other “Figure S2”.
Line 214: What do you mean by "relatively reliable", the term says that trust can be relative. Do you mean you may or may not be confident?
DISCUSSION
With the quality of results that present say, one would expect a more extensive discussion, especially by the amount of results shown. Indicate whether other studies have shown the same validation percentages, for example. They discuss only two genes, out of the 8 they indicated as candidate genes.
Lines 241-244: Is it necessary to repeat this ??
You can include the figure or table you refer to in the discussion. Example for the sentence in lines 249-251: insert at the end of the sentence (Figure 1a).
260-261: Readers at this stage of the work already know what the study is about. You don't have to repeat it.
CONCLUSION
Lines 282-284 The first sentence is the objective of the work, is it necessary to repeat at the conclusion? They can start with the phrase from Lines 284-285 Example: “The“ muscle contraction ”pathway is significantly enriched by both the DEGs and DAS genes ……”
Lines 287-288 The last sentence can be direct: “This study provides valuable resources for further research on the genetic and biological basis of heterosis in mule.”
Author Response
Reviewer
The study is interesting, it is the continuation of another study of the same authors. Results show that despite expected on the heterosis of Mules and Hinnies, there is not yet published in the literature on the candidate genes presented here, paving the way for new studies. Suggested changes to accept the publication are as follows:
1. TITLE
The authors conclude that there are candidate genes (line 285), it may be indicated to put the title “candidates genes”.
Authors: Many thanks for your positive comments. We have carefully revised the manuscript following your suggestions.
The TITLE has been revised please see that.
2. INTRODUCTION
A good explanation of the topic under study, short and objective. Authors should pay attention to: Lines 42: The number of references to animals could be higher than that of Plants, since they refer to 2 and 7 references, respectively. Even though there are many more studies of plants, there are more than two studies of animals that can be cited.
Authors: It has been corrected. Please see P1:L46.
Lines 51: Avoiding the terms "Although the recent" with the evolution and the passage of time, is not recent. Include an analogy to the moment (decade x, around year x, beginning of century x).
Authors: It has been revised. Please see P3:L57.
3. MATERIAL AND METHODS
It allows the repetition of the study, the Pac-Bio are correctly inserted NCBI.
The number (N) of animals evaluated in the study, despite appearing in the study by the same authors in https://www.ncbi.nlm.nih.gov/pmc/articles/PMC6680129/ I think it should be inserted in the present study, obliging the reader to go to another study just to consult the N, it is not necessary.
Authors: We agree with the reviewer comment. Details about the number of animals and tissue samples have added in the “sample collection” section. Please see P4:L83-87.
Line 90: It is correct, but you can put the reference that indicates the use of the ideal P-value to be 0.05, when using the DESeq2 R package (v 1.10.1), or in the package manual, it indicates, if yes, you do not need to repeat the reference then.
Authors: It has been removed. Please see P5:L113.
4. RESULTS
Throughout the study, the authors repeat several times what they are assessing, will not be necessary. Example: Phrase from line 112-113.
Authors: Thanks for your comment! It has been removed.
Line 116: it will be interesting to discuss and compare these values in the discussion section.
Authors: Thanks for your comment. This is the first study that we have used different tissues (muscle, brain and skin) from crosses between donkeys and horses. we have added some details about identified genes in discussion section. Please see P15:L330-347.
Lines 134 - 140 and serves for this whole paragraph: Table 1 is missing an indication at the end of the first sentence and review other text and see if the tables and figures are missing as soon as they speak the first time in the result.
Authors: It has been removed. Please see P7:L161.
They do not need to repeat the numerical values of the results of the tables, just put the percentages in the text, nor do they need to indicate the percentage differences of both, if one is 48.9% (line 134) logically that the other is 51.1% (line 135). Example as they can write: “Among them 48.9%, 58.6% and 53.6% of DEGs exhibited up-regulated in the above three tissues, respectively, while the other DEGs exhibited down-regulated (Table 1).”
Authors: It has been revised. Please see P7:L159-198.
Tables S1 and S2 can be joined.
Authors: It has been revised
Line 146: Correct the indication for the supplementary figure “… and Supplementary Figure 2” keeping the pattern they placed in the other “Figure S2”.
Authors: Thanks for your comment! It has been corrected. Please see P7:L171.
Line 214: What do you mean by "relatively reliable", the term says that trust can be relative. Do you mean you may or may not be confident?
Authors: It has been corrected. Please see P11:L269.
5. DISCUSSION
With the quality of results that present say, one would expect a more extensive discussion, especially by the amount of results shown. Indicate whether other studies have shown the same validation percentages, for example. They discuss only two genes, out of the 8 they indicated as candidate genes.
Authors: We have added some details about the identified DEGs and DAS genes in “discussion section”, please see P15:L330-347.
Lines 241-244: Is it necessary to repeat this??
Authors: It has been removed.
You can include the figure or table you refer to in the discussion. Example for the sentence in lines 249-251: insert at the end of the sentence (Figure 1a). 260-261: Readers at this stage of the work already know what the study is about. You don't have to repeat it.
Authors: Thanks for your comment! It has been added. Please see P15:L316.
- CONCLUSION
Lines 282-284 The first sentence is the objective of the work, is it necessary to repeat at the conclusion? They can start with the phrase from Lines 284-285 Example: “The“ muscle contraction ”pathway is significantly enriched by both the DEGs and DAS genes ……”
Authors: Thanks for your comment! It has been corrected. Please see P17:L370.
Lines 287-288 The last sentence can be direct: “This study provides valuable resources for further research on the genetic and biological basis of heterosis in mule.”
Authors: Thanks for your comment. It has been revised based on your suggestion. Please see P17:L373.
Reviewer 3 Report
I have reviewed the paper entitled "Comparative transcriptome profiling analysis uncovers novel heterosis-related genes associated with muscular endurance in mule". The paper is well written and the analyses are sounds and well described. I enjoyed reading it. I do not have specific comments, the only minor suggestion is to extend the discussion a bit more, for instance taking into account recent publications in sport horses where they found potential signs of selection for genes related to muscle contraction, synapse responses as well as neuronal system. I think it would be great to combine this study with those recent results as it might also help to understand better the genes important for sport performance in horses.
Author Response
Reviewer
I have reviewed the paper entitled "Comparative transcriptome profiling analysis uncovers novel heterosis-related genes associated with muscular endurance in mule". The paper is well written and the analyses are sounds and well described. I enjoyed reading it. I do not have specific comments, the only minor suggestion is to extend the discussion a bit more, for instance taking into account recent publications in sport horses where they found potential signs of selection for genes related to muscle contraction, synapse responses as well as neuronal system. I think it would be great to combine this study with those recent results as it might also help to understand better the genes important for sport performance in horses.
Authors: Thanks to the reviewer for this advice. We have added some information about recent horse genome studies in the discussion section, please see P16:L361-366.